# Biological Processes Highlighted in *Saccharomyces cerevisiae* during the Sparkling Wines Elaboration

**DOI:** 10.3390/microorganisms8081216

**Published:** 2020-08-11

**Authors:** María del Carmen González-Jiménez, Teresa García-Martínez, Anna Puig-Pujol, Fina Capdevila, Jaime Moreno-García, Juan Moreno, Juan Carlos Mauricio

**Affiliations:** 1Department of Microbiology, University of Cordoba, 14014 Cordoba, Spain; b02gojim@uco.es (M.d.C.G.-J.); b62mogaj@uco.es (J.M.-G.); mi1gamaj@uco.es (J.C.M.); 2Department of Enological Research, Institute of Agrifood Research and Technology—Catalan Institute of Vine and Wine (IRTA-INCAVI), Vilafranca del Penedès, 08720 Barcelona, Spain; anna.puig@irta.cat (A.P.-P.); fcapdevila@gencat.cat (F.C.); 3Department of Agricultural Chemistry, University of Cordoba, 14014 Cordoba, Spain; qe1movij@uco.es

**Keywords:** *Saccharomyces cerevisiae*, sparkling wine, fermentation, GO terms, protein

## Abstract

Sparkling wines elaboration has been studied by several research groups, but this is the first report on analysis of biological processes according to the Gene Ontology terms (GO terms) and related to proteins expressed by yeast cells during the second fermentation of sparkling wines. This work provides a comprehensive study of the most relevant biological processes in *Saccharomyces cerevisiae* P29, a sparkling wine strain, during the second fermentation under two conditions (without and with endogenous CO_2_ overpressure) in the middle and the end of second fermentation. Consequently, a proteomic analysis with the OFFGEL fractionator and protein identification with LTQ Orbitrap XL coupled to HPLC were performed. The classification of biological processes was carried out using the tools provided by the *Saccharomyces* Genome Database. Results indicate that a greater number of biological processes were identified under condition without CO_2_ overpressure and in the middle of the fermentation versus the end of the second fermentation. The biological processes highlighted under condition without CO_2_ overpressure in the middle of the fermentation were involved in the carbohydrate and lipid metabolic processes and catabolic and biosynthetic processes. However, under CO_2_ overpressure, specific protein expression in response to stress, transport, translation, and chromosome organization and specific processes were not found. At the end of fermentation, there were higher specific processes under condition without CO_2_ overpressure; most were related to cell division, growth, biosynthetic process, and gene transcription resulting in increased cell viability in this condition. Under CO_2_ overpressure condition, the most representative processes were related to translation as tRNA metabolic process, chromosome organization, mRNA processing, ribosome biogenesis, and ribonucleoprotein complex assembly, probably in response to the stress caused by the hard fermentation conditions. Therefore, a broader knowledge of the adaptation of the yeast, and its behavior under typical conditions to produce sparkling wine, might improve and favor the wine industry and the selection of yeast for obtaining a high-quality wine.

## 1. Introduction

The production of sparkling wine by the traditional, or *Champenoise,* method consists of two consecutive fermentations and a long period of aging of the wine in contact with the yeast lees. A first fermentation occurs, where the base wine is obtained from grape must, and a second fermentation of the base wine mixed with sugar and specific yeast strains of *Saccharomyces* occurs in a sealed bottle. After the second fermentation, the wine is subjected to a long aging period in contact with the yeast lees for at least 9 months (Spanish sparkling wine, Cava) at low temperatures (12‒16 °C). During this last stage, the organoleptic properties of wine are influenced by the release of certain secondary metabolites, such as glycerol, acetate, succinate, pyruvate, and esters [1,2,3,4]. Numerous studies attempt to relate the change in the concentration of aromatic compounds produced during aging and their effect on the final aroma of sparkling wines [5,6,7,8,9]. A recent study, conducted by our research group, focused on the study of the effect of endogenous CO_2_ overpressure during the second fermentation in the production of sparkling wines. Regarding volatilome and its relationship with the organoleptic properties of the final wine, CO_2_ overpressure affected the concentration of certain aromatic compounds, such as ethyl dodecanoate, ethyl tetradecanoate, hexyl acetate, ethyl butanoate, and ethyl isobutanoate [10]. Once yeast cells die during aging, intracellular compounds are released to the wine as mannoproteins during the autolysis process, thus improving the quality, and favoring the organoleptic properties of these special wines [11,12]. During these stages, the yeast is subjected to various stresses, so ensuring its survival and tolerance to these conditions is essential to guarantee the success of the sparkling winemaking process [13]. On the other hand, during yeast selection, early cell death and autolysis are wanted to accelerate the process of elaboration of sparkling wines.During the second fermentation, yeasts are subjected to various stress factors, such as high ethanol content, nitrogen deficiency, low pH values, low temperature, and CO_2_ overpressure [14]. Penacho et al. [14] showed that genes related to aerobic respiration, and vacuolar and peroxisomal functions are the main pathways affected by the pressure conditions, pointing to ethanol as the main factor responsible for these changes during second fermentation [14]. Furthermore, previous research reported that CO_2_ overpressure seems to affect the yeast stress-related proteome during the second fermentation. In fact, according to these authors, proteins involved in osmotic stress, such as glycerol biosynthesis and others required for energy storage mechanisms during nutrient starvation, stand out under second fermentation conditions [15].

Since there are few researches focused on analysis of biological processes according to the Gene Ontology (GO) terms under fermentation conditions, the present work aims to study the effect of endogenous CO_2_ overpressure, released by *Saccharomyces cerevisiae* P29 strain, on the proteins related to the GO terms of sparkling wines elaborated according to the *Champenoise* method. This work conducts a global study about the most relevant biological processes in *S. cerevisiae* P29, a sparkling wine strain, under two conditions: a control condition without pressure, or P (−), and a pressure condition, or P (+), at the middle (MF) and the end (EF) of fermentation. We designed a proteomic analysis based on an OFFGEL fractionation, and later protein identification (LTQ Orbitrap XL mass spectrometer coupled to HPLC) and selection using the Gene Ontology tools from the *Saccharomyces* Genome Database (SGD). Finally, the GO Term Mapping tool was used to determine the most significant biological processes and pathways found under both conditions of this study. Broader knowledge about the adaptation of the yeast and its behavior under the typical conditions of the second fermentation would improve the industrial process, and favor the selection and genetic improvement of wine yeast strains capable of carrying out this process, giving rise to a wine of quality.

## 2. Materials and Methods

### 2.1. Microorganism, Cultivation Conditions, and Sampling

For this work, the P29 strain (CECT 11770) of *Saccharomyces cerevisiae* was used. This is a typical sparkling wine strain for secondary fermentation in a closed bottle in the Cava production. It was isolated from the designation of origin (DO) of the Penedès, Barcelona (Spain) [16].

Yeast strain was incubated for 5 days at 21 °C, using gentle agitation of 100 rpm for growth in a pasteurized must of the Macabeo grape variety (174.9 g/L of sugar, 18.5° Brix, 3.6 g/L of total acidity, and 3.43 pH). When an ethanol content of 10.39% (*v*/*v*) was reached, yeast cells were introduced into the bottles together with the commercial base wine composed of Macabeo:Chardonnay (6:4), an ethanol content of 10.21% (*v*/*v*), 22 g/L sucrose, and 1.5 × 10^6^ cells/mL. The second fermentation was carried out in a thermostat chamber at 14 °C in bottles with a volume of 750 mL.

The bottles were divided into two groups for the study of CO_2_ overpressure. Half of the bottles were closed with a perforated shutter, without pressure condition (P (−)). The rest of the bottles were hermetically sealed with a shutter and with a metal crown capsule, this causes the CO_2_ released during the second fermentation to be trapped inside the bottle, with pressure condition (P (+)). During the second fermentation in the sparkling wine process, two sample times were taken: (a) at the middle of the second fermentation (MF) (pressure of 3 bar); and (b) at the end of the second fermentation (EF) (pressure of 6.5 bar) under CO_2_ overpressure condition. At the same time, when a similar consumed sugar and produced ethanol content was obtained under both conditions (sugar consumption (MF: 9.07 ± 0.26 g/L and EF: 0.3 ± 0.0 g/L) and ethanol content (MF: 10.7 ± 0.03% *v*/*v* and EF: 11.56 ± 0.04% *v*/*v*)), samples were taken from the condition without CO_2_ pressure. 

The wine compositions at the different sampling times were: (a) under MFP (−): 10.64 ± 0.02% *v*/*v* ethanol, 9.9 ± 0.5 g/L reducing sugars, 0.2 ± 0.0 g/L volatile acidity, 5.2 ± 0.1 g/L total acidity, 3.28 ± 0.03 pH, 1.87 ± 0.06 g/L malic acid, 0.1 ± 0.0 g/L lactic acid; (b) under EFP(−): 11.52 ± 0.03% *v*/*v* ethanol, 0.3 ± 0.0 g/L reducing sugars, 0.24 ± 0.03 g/L volatile acidity, 5.1 ± 0.02 g/L total acidity, 3.28 ± 0.01 pH, 1.90 ± 0.06 g/L malic acid, 0.1 ± 0.0 g/L lactic acid; (c) under MFP(+): 10.85 ± 0.04% *v*/*v* ethanol, 8.23 ± 0.06 g/L reducing sugars, 0.2 ± 0.0 g/L volatile acidity, 5.2 ± 0.0 g/L total acidity, 3.28 ± 0.01 pH, 1.87 ± 0.06 g/L malic acid, 0.1 ± 0.0 g/L lactic acid; (d) under EFP(+): 11.56 ± 0.04% *v*/*v* ethanol, 0.3 ± 0.0 g/L reducing sugars, 0.23 ± 0.02 g/L volatile acidity, 5.25 ± 0.02 g/L total acidity, 3.3 ± 0.02 pH, 1.89 ± 0.06 g/L malic acid, 0.1 ± 0.0 g/L lactic acid.

### 2.2. Viability

The cellular viability counting was carried out using appropriate dilutions with Ringer’s solution. Then, these were plated in Sabouraud agar medium for 48 h at 28 °C. All samples were carried out in triplicate. Data are recorded as means (± standard deviation) CFU (colony-forming units)/mL.

### 2.3. Proteomic Analysis

The cells were collected from all bottles by centrifugation at 4500× *g* for 10 min by a centrifuge (Rotina-38, Kirchlengern, Germany), washed the sediment twice with sterile distilled cold water and broken through a mechanical technique in Vibrogen, using the Cell Mill V6 (Edmund Bühler, Bodelshausen, Germany) with 500 µm diameter glass balls. Once the cells were broken, protein extraction was performed. For this purpose, a protease inhibitor cocktail and an extraction buffer composed of 100 mM Tris-HCl (pH 8), 0.1 mM ethylenediaminetetraacetic acid (EDTA), 2 mM dithiothreitol (DTT), and 1 mM phenylmethylsulfonyl fluoride (PMSF) were used. A total of 500 µg of protein was loaded into the well tray of the protein fractionator. This concentration was calculated by a Bradford test [17]. Then, the proteins were separated according to their isoelectric point, using the OFFGEL 3100 fractionator from Agilent Technologies (Alto Palo, CA, USA). For this, the protein samples were solubilized in a Protein OFFGEL fractionation buffer containing urea, thiourea, DTT, glycerol, and buffer with ampholytes. For identification, protein samples were analyzed on an LTQ Orbitrap XL (Thermo Fisher Scientific, San Jose, CA, USA) mass spectrometer equipped with a nano LC Ultimate 3000 system (Dionex, Germering, Germany), accompanied by the Proteome Discoverer 1.0 database (United States) at the University’s Central Research Support Service (SCAI) from Córdoba, Spain. Conditions are described in more detail in articles published by the research group [18].

### 2.4. Biological Processes Analysis

To carry out a more reliable selection of the total proteins, the following selection criteria were taken into account, according to Dasari et al. [19]: (i) score greater than 2; (ii) number of peptides observed greater than or equal to 2. The resulting proteins were classified into biological processes, according to the GO terminology of the *Saccharomyces* Genome Database (SGD), using the GO Slim Mapper tool.

The proteins involved in each biological process were classified in different GO Terms provided by the SGD database (https://www.yeastgenome.org/). This classification was made using the GO Term Finder tool, provided by the database. The GO Terms are descriptive terms that allow the relating of each gene product with a molecular, cellular, and biological process context, providing a statistical value (*p*-value). This statistical study was performed with a level of significance (α) of 0.01.

## 3. Results and Discussion

The present study was performed on cultures of *S. cerevisiae* P29 strain under two experimental conditions: without and with CO_2_ overpressure at the middle and the end of fermentation under real second fermentation conditions (Figure 1).

A total of 1517 proteins were detected under the MFP (−) and 594 under MFP (+); 542 of them were common under both conditions (viz., 975 proteins specifically detected in MFP (−) and 52 under MFP (+)). On the other hand, 392 proteins were identified under EFP (−) and 419 under EFP (+), with 268 proteins being common between both conditions (viz., 124 proteins were specific in EFP (−) and 150 under EFP (+)) (Appendix A).

The biological processes obtained in each type of condition and sampling times were elucidated from the GO term of biological processes and pathways. Gene Ontology is a tool that describes how and where gene products work in biological systems. It is structured in three interrelated parts that describe what genetic products do at the biochemical level and at the cellular level, and the general biological objectives to which their actions contribute [20].

### 3.1. Biological Processes during the Middle of the Second Fermentation

A total of 57 biological processes were obtained in the middle of the second fermentation (Figure 2). While 18 of them were specific processes under the condition without overpressure, there were no specific processes detected under CO_2_ overpressure condition. This difference could be because, in high-stress situations, the yeast cells only carry on those biological processes that are fundamental to viability and cell maintenance. These results agree with those from Matallana et al. [21], which also indicates that the tolerance to environmental stress conditions is a key factor to achieve the biotechnological success of *Saccharomyces* yeasts.

On the other hand, there were 18 specific processes under MFP (−); those processes that showed a higher protein frequency were related to metabolism, and more specifically to the metabolism of lipids and carbohydrates (6.49% and 5.97%, respectively). Both processes were related to a series of GO Terms (data shown in Appendix A) among which the biosynthesis of lipids such as sterols, glycerolipids, or phospholipids stood out. The yeast plasmatic membrane can play an important role in the transport and tolerance of toxic compounds such as ethanol, acetic acid, acetaldehyde, and medium-chain fatty acids, and therefore small changes in the plasma membrane could trigger a modification of the yeast metabolism [22,23]. Furthermore, sterols are involved in membrane dynamics, controlling lateral movements and the activity of membrane proteins [24]; while glycerolipids play an important role in cell signaling, membrane trafficking, and anchoring of membrane proteins [25]. The composition of fatty acids in the plasma membrane of yeasts depends on the lipid composition of the environment, the availability of oxygen, and the fermentation conditions. When fermentation takes place under anaerobic conditions, the yeast can synthesize more saturated fatty acids and less ergosterol and squalene [24]. However, when fermentation carries on under aerobic conditions, fewer medium-chain fatty acids and more unsaturated fatty acids are produced [26]. This causes a change in the composition of its lipid fractions by reducing the surface area of the membranes and decreasing the viability of the cells during fermentation. These differences in lipid composition between the two conditions are due to the fact that the biosynthetic pathways of these essential lipids are strictly aerobic [27] and are related to mitochondrial activity [28]. However, for validation of obtained results, oxygen consumption measurements by yeast cells should be made throughout the second fermentation in bottle, and analysis of the lipid composition of the plasma membranes under both conditions must be performed. On the other hand, the main GO Terms related to “carbohydrate metabolism” were metabolism of glucans and glycogen (data shown in Appendix A). These polysaccharides, more specifically β-glucans, provide rigidity to the cell wall. The synthesis of these polysaccharides is involved in the yeast budding process. In addition, the metabolism of glycogen, a reserve polysaccharide, is also involved in increasing the viability of yeast cells in wine fermentations [29,30]. Both processes (lipid and carbohydrate metabolism) are related to the production of energy through glycolysis, the pentose phosphate pathway, and the TCA cycle. This energy is used by the yeast for cell growth, morphogenesis, and biosynthesis of compounds such as amino acids or proteins. The rest of the specific biological processes identified under MFP (−) presented a protein frequency between 2 and 0.1% (Figure 2). These results were in contrast with the cell viability analysis: under without overpressure 5.53 × 10^6^ (±2.1 × 10^6^) CFU/mL, (10.64 ± 0.02% *v*/*v* ethanol) and under CO_2_ overpressure 8.02 × 10^6^ (±1.4 × 10^6^) CFU/mL (3 bar, semi-sparkling wine, 10.85 ± 0.04% *v*/*v* ethanol), no significant differences were observed [31]. This suggests that the fundamental processes of cell growth and those related to the central metabolism of yeast would be taking place under both conditions, and in this case, the CO_2_ overpressure would not affect cell viability, despite being subjected to stress.

With the aim of contrasting this hypothesis, the common biological processes at the middle of the second fermentation were identified (39 processes), and the most representative common biological processes under MFP (−) versus MFP (+) were eight processes (biosynthetic process, catabolic process, cell wall organization or biogenesis, cellular nitrogen compound metabolic process, cofactor metabolic process, cytoskeleton organization, small molecule metabolic process, and sulfur compound metabolic process). In view of these results, the yeast could have a more active metabolism under conditions without CO_2_ overpressure versus CO_2_ overpressure, since the highlighted processes are related to synthesis and degradation, presumably of lipids and carbohydrates (specific processes identified), and to obtaining energy (cofactor metabolic process, cellular nitrogen compound metabolic process, or small molecule metabolic process). Unlike yeasts under CO_2_ overpressure condition, they dedicate most of their energy to their survival. Processes that were differentially up-regulated (relative frequency above twofold) under MFP (+) versus MFP (−) samples were as follows:chromosome organization, translation, nucleocytoplasmic transport, response to stress, ribosome biogenesis, anatomical structure formation involved in morphogenesis, cell differentiation, chromosome segregation, membrane organization, DNA metabolic process, homeostatic process, protein targeting, aging, anatomical structure development, and cellular protein modification process. These processes are related to cell division (chromosome organization: 14.29%; chromosome segregation: 8.16%; DNA metabolic process: 6.12%), morphogenesis (anatomical structure formation involved in morphogenesis: 8.16%; anatomical structure development: 2.04%; membrane organization: 8.16%) and gene transcription (translation, 14.29%; nucleocytoplasmic transport, 10.2%; ribosome biogenesis, 10.2%; cellular protein modification process, 2.04%). In view of these processes under MFP (+), yeast metabolism under CO_2_ overpressure conditions and its cell growth are not affected, since it counteracts not having specific processes for them by presenting a high frequency of proteins in the common processes mentioned above. However, the yeast could be stressed by CO_2_ pressure, since a stress response under MFP (+) higher than MFP (−) has been detected.

### 3.2. Biological Processes at the End of the Second Fermentation

A total of 46 processes were obtained under condition without CO_2_ overpressure (EFP (−)), while under condition of CO_2_ overpressure (EFP (+)) 42 biological processes were identified at the end of the second fermentation (Figure 3). Six of them were specific under condition EFP (−) and two in the EFP (+).

Specific biological processes in EFP (−) were aging (4.03%), cell division (2.42%), cell morphogenesis (1.61%), cytoskeleton-dependent intracellular transport (0.81%), growth (2.42%), and protein maturation (2.42%). In both conditions, due to the scarcity of the carbon source (glucose), the yeast reorganizes its metabolism during the stationary phase [29,30]. The viability of yeast cells under EFP (−) condition was higher than under condition with CO_2_ overpressure; this is in accordance with the identification of a greater number of biological processes related to cell growth and division (EFP (−): 1.14 × 10^6^ (±0.4 × 10^6^) CFU/mL (11.52 ± 0.03% *v*/*v* ethanol); EFP (+): 3.33 × 10^4^ (±1.2 × 10^4^) CFU/mL (11.56 ± 0.04% *v*/*v* ethanol). This fact may be explained since, as described above, in EFP (+), yeast autolysis may be taking place, which would help the survival of the rest of the population in this hard condition. On the other hand, the GO Terms related to the specific biological processes in EFP (−) were developmental process, intracellular copper ion transport, and oxidoreduction coenzyme metabolic process (Appendix A). They presented *p*-values of 0.00365, 0.00488, and 0.00738, respectively. These GO Terms could be involved with the maintenance of the redox balance between NAD^+^ and NADH in aerobic conditions by *S. cerevisiae*. Yeasts need to recycle NAD^+^ and oxidize NADH for the continuation of glycolysis; otherwise, the glycolytic flow would decrease, which could lead to a depletion of the ATP energy charge, becoming lethal for yeast [31,32]. According to Kutyna et al. [33], most of the NADH produced during glycolysis is subsequently oxidized during the formation of ethanol. NAD^+^ regeneration can also occur through the formation of glycerol. Peeters et al. [34] reported that the combination of aberrant growth onset due to glycolytic dysregulation in cells can cause apoptosis in yeast.

Regarding common biological processes, under condition without CO_2_ overpressure (EFP (−)) processes which stood out were biosynthetic process, cellular amino acid metabolic process, small molecule metabolic process, carbohydrate metabolic process, cofactor metabolic process, generation of precursor metabolites and energy, signal transduction, homeostatic process, cell death, cell wall organization or biogenesis, cellular protein modification process, and lipid metabolic process.In view of the results obtained, most of these common processes in EFP (−) are involved in maintaining the redox balance caused by oxidative stress. For this purpose, the yeast could be synthesizing amino acids, such as methionine or tryptophan. These amino acids are described in studies carried out by Peláez-Soto et al. [35] as cellular protectors against damage caused by the oxidative metabolism of yeast. In addition, the processes of carbohydrate metabolic, cofactor metabolic, generation of precursor metabolites and energy, signal translation, and lipid metabolic would be dedicated to the production of energy needed to induce the antioxidative machinery. At last, seven processes presented a frequency more than twofold under condition with CO_2_ overpressure versus without CO_2_ overpressure, and the majority of these processes are related to gene translation (translation, ribonucleoprotein complex assembly, ribosome biogenesis, nucleocytoplasmic transport, mRNA processing, and transmembrane transport). The most significant GO Terms obtained for these processes coincide with the biological processes previously mentioned. The biogenesis of ribosomes (*p*-value: 1.35 × 10^−8^) and protein translation (*p*-value: 2.57 × 10^−6^) stood out. Presumably, the yeast could be synthesizing stress response proteins, and is more likely to respond to stress due to CO_2_ overpressure, since the conditions of ethanol and glucose did not show significant differences between conditions (see Materials and Methods section).

The specific biological processes under EFP (+) were tRNA metabolic process and chromosome organization, which presented a frequency of 1.99% and 3.97%, respectively. Four GO Terms have been related to these two specific biological processes (Appendix A). These GO Terms were chromosome organization (*p*-value: 0.00441), DNA packaging (*p*-value: 0.00909), DNA-templated transcriptional start site selection (*p*-value: 0.00991), and transcriptional start site selection at RNA polymerase II promoter (*p*-value: 0.00991). These results suggest that yeast could be activating its transcription and translation machinery to try to cope with cell death and autolysis. Jing et al. [36] reported that the increase in the amount of ethanol in the medium induces the process of autophagy in the yeast during fermentation. Yeast autolysis involves the release of different products, resulting from the degradation of yeast macromolecules, into the wine. In *S. cerevisiae*, this process is triggered by starvation conditions, and is considered an adaptive response, allowing the survival of younger yeast cells, due to the recycling of the compounds released into the medium by dead yeasts [37]. 

## 4. Conclusions

This work focuses on providing a first approach to an understanding of the biological processes that take place in yeasts during the second fermentation in the production of sparkling wines through a proteomic study. This knowledge arises from the need to select yeasts capable of carrying out the second fermentation, and of tolerating various stresses to which they are subjected. We believe that stress tolerance is a good criterion for selecting enologically interesting yeasts. Despite obtaining a greater number of biological processes in the middle of the second fermentation, no specific process of the condition with CO_2_ overpressure was identified. However, at the end of the second fermentation, two specific processes took place under CO_2_ overpressure, versus six specific processes under conditions without CO_2_ overpressure. Most of the prominent processes were related to cell division, growth, and biosynthetic process, presumably for the maintenance of good cell viability observed without CO_2_ overpressure condition. At the end of fermentation, it was clear that the overpressure affects viability, since there were a smaller number of living cells in the second fermentation under this condition. However, under CO_2_ overpressure condition, processes were related to tRNA metabolic process, chromosome organization, and translation.

These results represent a first approach that highlights the most relevant biological processes that take place during the second fermentation in bottle in the production of Spanish sparkling wine (Cava). Furthermore, they couldbe a starting point for future research to validate some of the possible mechanisms that take place during the second fermentation in bottle.

A better understanding of yeast metabolism, its adaptation, and its behavior under the typical conditions of the second fermentation is necessary for the strain selection process and improvement of its application in the wine industry, and can be interesting for quality control and improvement of the sparkling winemaking process.

## Figures and Tables

**Figure 1 microorganisms-08-01216-f001:**
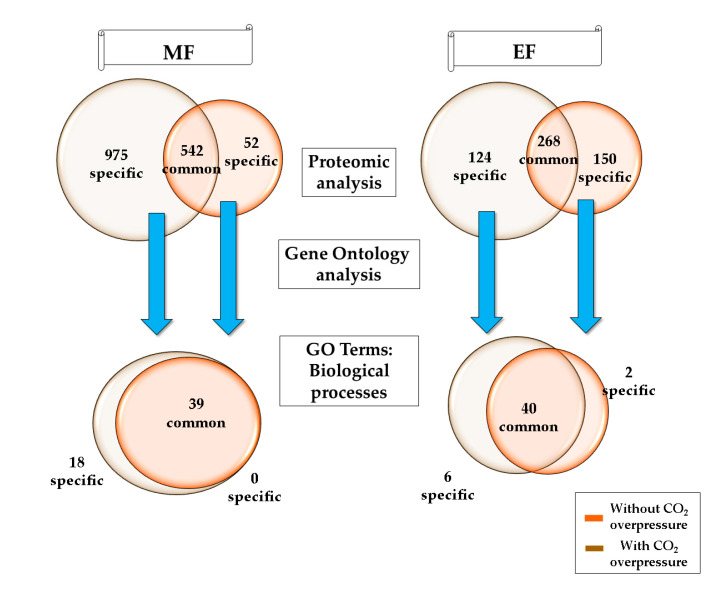
Venn diagram of proteins and Gene Ontology (GO) terms related to the biological processes under two experimental conditions: without and with CO_2_ overpressure at the middle (MFP (−) and MFP (+), respectively) and the end of fermentation under real second fermentation conditions (EFP (−) and EFP (+), respectively).

**Figure 2 microorganisms-08-01216-f002:**
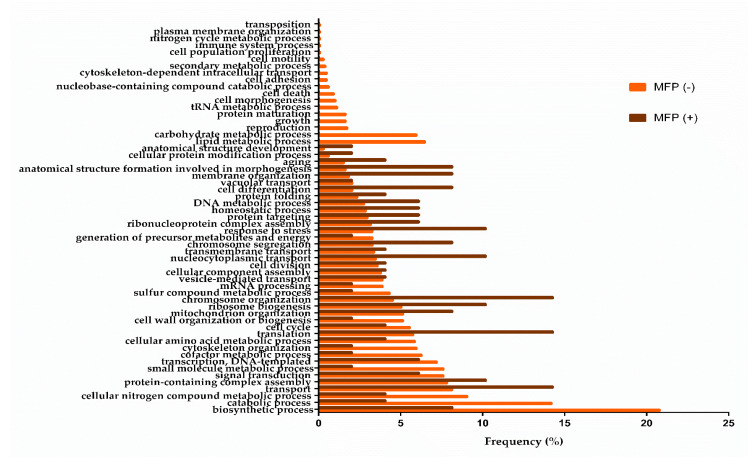
Relevant biological processes in middle of second fermentation without CO_2_ overpressure (MFP (−) in orange), and with overpressure (MFP (+) in brown). The frequency (%) of each process corresponds to the number of proteins involved in each process with respect to the total proteins identified.

**Figure 3 microorganisms-08-01216-f003:**
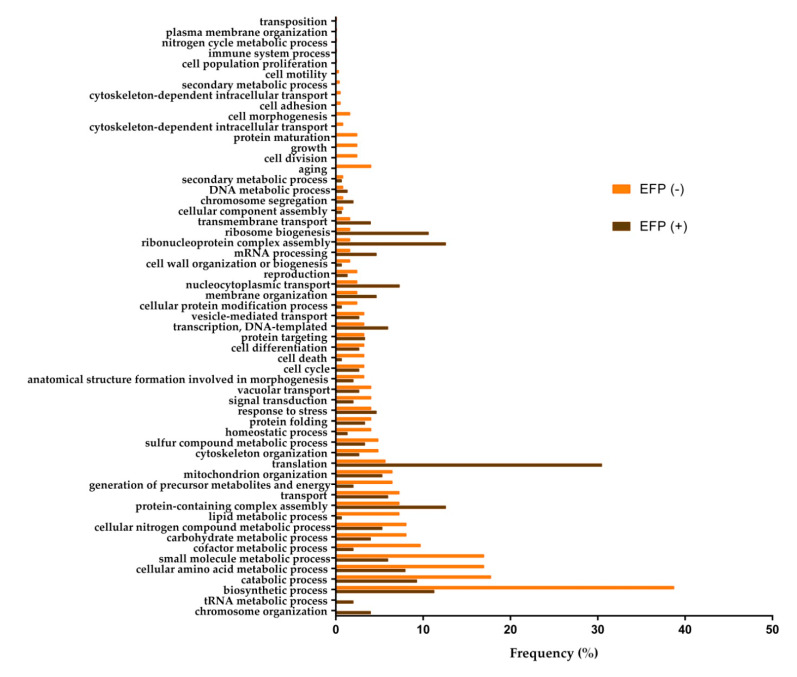
Relevant biological processes at the end of the second fermentation without CO_2_ overpressure (EFP (−) in orange), and with overpressure (EFP (+) in brown). The frequency (%) of each process corresponds to the number of proteins involved in each process with respect to the total proteins identified.

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
