# Peer review of "Biological Processes Highlighted in Saccharomyces cerevisiae during the Sparkling Wines Elaboration"

_microorganisms, 2020, doi:10.3390/microorganisms8081216_

Round 1

Reviewer 1 Report

The authors have clearly improved the manuscript and have satisfactorily dealt with the comments raised by this reviewer.

Author Response

We appreciate your comment.

Reviewer 2 Report

The present study provides GO terms of wine yeast depending on some conditions. This is quite interesting for me because GO term has been widely used for interpreting or categorizing sets of various genes. But I doubt that only GO term data can sufficiently demonstrate the relationship between biological processing and different conditions such as CO2 overpressure and sampling point. I think the authors should add some more data to be accepted in this journal.

These are major and minor suggestions:

  1. Line 57-60 and line 61-64 are somewhat duplicated. Revise it.
  2. Line 89 – provide a reference for strains used in this study.
  3. Line 100 – can you specify CO2 pressure condition?
  4. Line 101-102 – I think different CO2 overpressure conditions lead different second fermentation time based on GO terms. Can you explain more specific sampling time?
  5. Line 197, 198, 238 – Viability can be misread because multiplication comes earlier than addition. I recommend you to use Log CFU/mL or bracket.
  6. Line 168, 222, 242, and others - Clarify supplementary table 1,2 or 3 for each data in manuscript.
  7. Why the authors didn’t provide wine properties at sampling points? Because there is a few information to explain GO terms depending on different conditions
  8. Can you provide metabolomics data?
  9. Can you explain the effect of CO2 overpressure on cell viability? Because cell viability of wine yeast under CO2 overpressure was higher at mid-phase of second fermentation, while that was lower at the end of second fermentation. If you can provide alcoholic fermentation trend, it will be helpful for understanding.

Author Response

Reviewer 2

The present study provides GO terms of wine yeast depending on some conditions. This is quite interesting for me because GO term has been widely used for interpreting or categorizing sets of various genes. But I doubt that only GO term data can sufficiently demonstrate the relationship between biological processing and different conditions such as CO2 overpressure and sampling point. I think the authors should add some more data to be accepted in this journal.

The Gene Ontology (GO) is a very useful tool that provides a system for hierarchically classifying genes or gene products into terms organized in a graph structure. The terms are groups into three categories: molecular function (describing the molecular activity of a gene), biological process (describing the larger cellular or physiological role carried out by the gene, coordinated with other genes) and cellular component (describing the location in the cell where the gene product executes its function).

This study is the first report on analysis of biological processes according to the Gene Ontology terms (GO terms) and related to proteins expressed by yeast cells during the second fermentation of sparkling wines.

As this is a broader work, it was decided to start with the general study of biological processes and then to focus on these different processes to obtain a wider knowledge and complete it with the analysis of the metabolome and the transcriptome. However, to achieve this study in its entirety, it is previously necessary to screen the different biological processes to focus these studies.

These are major and minor suggestions:

Line 57-60 and line 61-64 are somewhat duplicated. Revise it.

This paragraph has been corrected.

Line 89 – provide a reference for strains used in this study.

A reference has been added: [16]

Line 100 – can you specify CO2 pressure condition?

The production of sparkling wine by the traditional method involves two consecutive fermentations. The second fermentation takes place in a closed bottle so that the released CO2 is trapped inside the bottle. This is called CO2 overpressure condition P (+).

This information has been clarified in the revised manuscript.

Line 101-102 – I think different CO2 overpressure conditions lead different second fermentation time based on GO terms. Can you explain more specific sampling time?

Under CO2 overpressure condition, the samples were taken in the middle of the second fermentation, MF (pressure of 3 bar, 7 days); and at the end of the second fermentation, EF (6.5 bar, 53 days). At the same time, when a similar consumed sugar and produced ethanol content was obtained in both conditions [sugar consumption (MF: 9.07 ± 0.26 g/L and EF: 0.3 ±0.0 g/L) and ethanol content (MF: 10.7±4 0.03 % v/v and EF: 11.56± 0.04 % v/v)] samples were taken from the condition without CO2 pressure.

This information has been added and completed in the new manuscript.

Line 197, 198, 238 – Viability can be misread because multiplication comes earlier than addition. I recommend you to use Log CFU/mL or bracket.

Viability data are recorded as means (± Standard Deviation) CFU (colony-forming units)/mL.

This has been modified in the manuscript.

Line 168, 222, 242, and others - Clarify supplementary table 1,2 or 3 for each data in manuscript.

This information has been added.

Why the authors didn’t provide wine properties at sampling points? Because there is a few information to explain GO terms depending on different conditions

The wine composition at the different sampling times was: a) under MFP (-) : 10.64 ± 0.02% v/v ethanol, 9.9 ± 0.5 g/L reducing sugars, 0.2 ± 0.0 g/L volatile acidity, 5.2 ± 0.1 g/L total acidity, 3.28 ± 0.03 pH, 1.87 ± 0.06 g/L malic acid, 0.1 ± 0.0 g/L lactic acid; b) under EFP(-): 11.52 ± 0.03% v/v ethanol, 0.3 ± 0.0 g/L reducing sugars, 0.24 ± 0.03 g/L volatile acidity, 5.1 ± 0.02 g/L total acidity, 3.28 ± 0.01 pH, 1.90 ± 0.06 g/L malic acid, 0.1 ± 0.0 g/L lactic acid; c) under MFP(+): 10.85 ± 0.04% v/v ethanol, 8.23 ± 0.06 g/L reducing sugars, 0.2 ± 0.0 g/L volatile acidity, 5.2 ± 0.0 g/L total acidity, 3.28 ± 0.01 pH, 1.87 ± 0.06 g/L malic acid, 0.1 ± 0.0 g/L lactic acid; d) under EFP(+): 11.56 ± 0.04% v/v ethanol, 0.3 ± 0.0 g/L reducing sugars, 0.23 ± 0.02 g/L volatile acidity, 5.25 ± 0.02 g/L total acidity, 3.3 ± 0.02 pH, 1.89 ± 0.06 g/L malic acid, 0.1 ± 0.0 g/L lactic acid.

This information has been added.

Can you provide metabolomics data?

This work forms part of a more complete research project carried out by our multidisciplinary research group, which has been divided into different research areas to provide a more exhaustive knowledge of the metabolism of Saccharomyces cerevisiae during the second fermentation in the production of sparkling wine. Some members of this research group has been in charge of studying the S. cerevisiae metabolome, these data have been published by Martínez-García et al., (2017): Reference [10] Martínez-García, R., García-Martínez, T., Puig-Pujol, A., Mauricio, J. C., Moreno, J. (2017). Changes in sparkling wine aroma during the second fermentation under CO2 pressure in sealed bottle. Food chemistry, 237, 1030-1040.

Can you explain the effect of CO2 overpressure on cell viability? Because cell viability of wine yeast under CO2 overpressure was higher at mid-phase of second fermentation, while that was lower at the end of second fermentation. If you can provide alcoholic fermentation trend, it will be helpful for understanding.

We have added the results of ethanol concentration under the conditions studied in the manuscript. This parameter is very similar in each condition, so the difference in cell viability is not due to ethanol, probably it is a consequence to mechanisms of autolysis enhanced by CO2 overpression.

Round 2

Reviewer 2 Report

I accept this manuscript.

This manuscript is a resubmission of an earlier submission. The following is a list of the peer review reports and author responses from that submission.

Round 1

Reviewer 1 Report

This paper reports the analysis of the major biological processes that take place in S. cerevisiae cells during the second fermentation of sparkling wines. Authors performed wine fermentations with a selected indigenous S. cerevisiae strain that is currently used for sparkling wine elaboration, and pasteurized grape must. They performed quantitative proteomic analyses of S. cerevisiae cells during the second fermentation of wines and under two fermentation conditions: with and without CO2 overpressure. Two fermentation times were considered: middle fermentation and the end of fermentation. Authors carried out a comparative analysis of the obtained results, and they used the Gene Ontology (GO) section of the Saccharomyces genome database. GO is a reliable source of information on the functions of genes.

Regarding the results of the study, authors identified a total of 1569 different proteins in middle fermentation samples, and 542 proteins in samples taken at the end of fermentations. The functional analysis and discussion of results are very clarifying of the events that take place in the yeast cells in response to the stress of CO2 overpressure during the alcoholic fermentation process, and it should be pointed out that it takes place in presence of an additional stress agent: ethanol (10.7- 11.6 % v/v).

In my opinion this study shows an innovative strategy and results that provide insight into the molecular mechanisms that yeast cells utilize to survive under highly stringent conditions. An additional strength of this study is the high interest of the reported results for the wine industry, as well as for biotechnological applications, because they provide useful tools for selection of robust yeast strains. In my opinion, the major weakness of the study is that the manuscript requires editing of English language and style.

Some additional comments

Please, replace the proposition "in" by "under" in all the expressions meaning "under conditions" along the whole manuscript.

Please, replace the transitive verb "to highlight" by the intransitive verb "to stand out" whenever it is required in the manuscript.

Abstract

page 1

line 16: replace "relating to proteins... condition." by: "and related to proteins expressed by yeast cells during the second fermentation of sparkling wines."

lines 26-27: replace "However, in the CO2 overpressure condition were response to stress, ... were not found." by: "However, under CO2 overpressure, specific protein expression in response to stress, ... were not found."

line 33: replace "probably as response to stress to this hard fermentation condition." by " probably in response to the stress caused by the hard fermentation conditions"

line 35: replace " favor to the wine industry and to the yeast selection..:" by " favor the wine industry and the selection of yeast ..."

Introduction

page 2

line 49: replace "have focused" by "has focused"

line 59: reorder the sentence: "On the other hand, the early cell death and autolysis is wanted when selecting yeasts to accelerate the process..." so that it reads: "On the other hand, during yeast selection early cell death and autolysis are wanted to accelerate the process...".

line 61: replace "production of cava from Spain" by "production of Spanish cava".

lines 61-62: rewrite the sentence: "Yeast survival and the stress tolerance capacity results essential to guarantee success during the second fermentation, in addition to be an important feature during strain selection", so that it reads: "Yeast survival and stress tolerance become essential to guarantee success during the second fermentation, and they are yeast traits of relevance for strain selection."

line 63: replace "Under these conditions,.." by " During the second fermentation,...".

line 75: include "elaborated" so that it reads: "sparkling wines elaborated according to the Champenoise method."

line 82: replace " between the two study conditions" by " found under both conditions of this study."

Materials and Methods

page 3

line 109: replace: "protein extraction was extracted." by "protein extraction was performed."

Results and Discussion

Figure 3:

It should read: "Relevant biological processes at the end of the second fermentation ....".

page 7

lines 161-162: replace: "that are fundamental to be able to viability and cell maintenance." by " that are fundamental to viability and cell maintenance.

lines 181- 182: replace "These differences ...... are because the biosynthetic pathways..." by " These differences ....... are due to the fact that the biosynthetic pathways..."

line 188: replace "In addition, glycogen metabolism as a reserve polysaccharide is also involved...." by " In addition, the metabolism of glycogen, a reserve polysaccharide, is also involved,....."

line 195: please delete: "in the condition" so that it reads: "... and under CO2 overpressure 8.02 ±...

page 8

lines 210-122: rewrite "Regarding the processes that reported above 2‐fold the frequency in MFP (+) versus MFP (‐) were ....."

so that it reads: "Processes that were differentially up-regulated (relative frequency above 2-fold) in MFP (+) versus MFP (-) samples, were as follows: ..."

lines 220-225: Rewrite the last three sentences of the pragraph.

line 231: remove "Concerning" so that it reads: "Specific biological processes in EFP (‐) were: aging...."

line 237: include "explained" so that it reads: " This fact may be explained since, as described above.....

page 9

lines 277-279: remove the final words: "because......... achieve the survival of the yeast."

lines 292-294: rewrite, so that it reads: "However, at the end of the second fermentation two specific processes took place under CO2 overpressure, versus six specific processes under conditions without CO2 overpressure."

Reviewer 2 Report

The authors have used a very powerful proteomic technique to monitor the yeast cellular proteome along a second sparkling wine fermentation in bottle (Cava production). As control condition, the same second fermentation was carried out without CO2 overpressure. The proteins detected at middle and end fermentation were used to determine the significant Gene Ontology terms (GO terms). The objective is interesting and could be useful for wine industry. However, the results presented are very descriptive and the conclusions obtained turned out very speculative. The analysis of GO terms gives a general view of the most important biological processes, most of them common to any fermentative process. The authors should have gone a step forward and validate some of the possible mechanisms that are happening during second fermentation in bottle.

Other minor comments:

  • Lines 91-93. Why the adaptation of the inoculum for the second fermentation was carried out by fermenting a Macabeo grape must? Why did not the authors adapted by a traditional “pied de cuve”? This adaptation process can determine the further performance during the second fermentation in bottle.
  • Line 95. Why the base wine was not supplemented with sucrose instead of glucose, as it is done in the industry?
  • Lines 142-143. The numbers do not fit with the EF Venn diagram
  • The quality of the figures 2 and 3 should be improved
  • Lines 165-177. Lipid metabolism was one of the main differential biological process in MFP (-). The authors explained this result by a higher oxidative metabolism and higher oxygen availability in MFP (-). However, it could be just the opposite. An open system enables a quicker stripping of oxygen when the CO2 is produced. In the MFP (+), oxygen availability could be longer and allow a better plasma membrane composition
  • Lines 183-184. It is very risky to state this because the authors have not determined the plasma membrane composition